# Distribution of Order Parameter in Solids under High Pressure Torsion

**DOI:** 10.3390/ma15196970

**Published:** 2022-10-07

**Authors:** Petr Straumal, Yuri Zavorotnev, Leonid Metlov, Olga Popova

**Affiliations:** 1Baikov Institute of Metallurgy and Materials Science, Russian Academy of Sciences, 119334 Moscow, Russia; 2Department of Complicated Dynamic Systems, Donetsk Institute for Physics and Engineering Named after A.A. Galkin, 83114 Donetsk, Ukraine; 3Department of Nanophysics, Donetsk National University, 83001 Donetsk, Ukraine; 4Department of Management and Financial and Economic Security, Donetsk National Technical University, 43024 Lutsk, Ukraine

**Keywords:** high pressure torsion, phase transformations, Landau phenomenological theory, solids, order parameter

## Abstract

Severe plastic deformation (SPD) can lead to various phase transformations. High-pressure torsion (HPT) is one of the most important variants of SPD. In principle, HPT can continue almost indefinitely long, as long as the plungers are not destroyed. However, the number of defects in a material during HPT deformation cannot increase indefinitely. When the rate of defects production becomes equal to the rate of their annihilation, a steady state or dynamic equilibrium is reached. Unexplored is the issue of establishing equilibrium at the initial stage of plunger torsion, when there is an angular acceleration. The parameters of the steady state are described here using the idea of an order parameter in solids in the framework of Landau phenomenological theory and the Landau–Khalatnikov equation.

## 1. Introduction

The anisotropy of individual crystallites can be detrimental if their size is comparable with the dimensions of the whole parts. One can refine the grains in a material with the aid of severe plastic deformation (SPD). Together with grain refinement, SPD can also lead to various phase transformations [1,2,3]. The important feature of SPD processes is that the sample during SPD cannot break. Actually, even the word “deformation” itself acquires a paradoxical meaning during SPD, since, in fact, the shape of the sample remains unchanged. High-pressure torsion (HPT) is one of the most important variants of SPD. In HPT, a flat sample is located in a tight volume between two rotating plungers under applied high pressure [4]. The HPT can continue almost indefinitely long, that is, as long as the plungers themselves are not destroyed [5,6,7,8,9]. However, the accumulation of defects in a material during HPT deformation cannot continue indefinitely. Simultaneously with the production of defects, their relaxation or annihilation begins. When the rate of defects production becomes equal to the rate of their annihilation, a steady state or dynamic equilibrium is reached and mass transfer continues in the sample bulk, which is determined by the rheology of the material [10,11,12,13,14,15,16]. This is especially pronounced during HPT when the processes of dissolution and precipitation of the second phase compete with each other. In particular, this competition and the emergence of a steady state during HPT were experimentally studied in the Cu-Ag, Cu-Sn, Cu-Co, and Cu–Ni systems [17,18,19].

Thus, in Refs. [17,18,19], the behavior of the order parameter, concentration, and phase boundaries in two-component copper-based alloys was studied under the action of non-destructive HPT. In this case, the outer boundary of the sample was fixed. It turned out that, regardless of the initial conditions (like annealing temperature), the system came to the same stationary state after a certain time (for copper-based alloys this is usually 1–2 plunger turns). The constructed theoretical models [20,21,22] made it possible to explain the observed phenomena. On the basis of the version of the theory [20,21], the question of the formation of a traveling wave was considered and its profile (kink) was calculated [23]. However, the question of the radial distribution of the order parameter, which is a linear combination of shifts of unit cell atoms, has remained unexplored, provided that the outer boundary of the sample is not fixed. This paper is devoted to the consideration of this problem in the cases of an ideal sample with one defect.

This problem arises when studying the distribution of stresses during the acceleration of the fan blades, centrifugal pump, etc. In particular, it was shown in [24] that when the blade of a centrifugal pump rotates, the pressure on the surface of the magnetic fluid sealing film has a periodic pulsation. This pressure is transferred to the blades, which leads to a change in the distribution of stress and order parameter (OP) on them. In addition, this issue is relevant in the study of voltage distribution in power networks in the event of peak loads [25,26]. In order to ensure the emission reduction in the process of extracting energy from fuel, one carries out the reactions far from stoichiometric conditions. The practical implementation of this requires a search for a compromise between emission, combustion stability, and efficiency and is associated with the swirling of the flame and the use of vortex technologies based on novel algorythms [27]. For example, the finite control set model predictive control strategy (FCS-MPC) with fast response and a simple and flexible structure was used for the first time to investigate the FCS-MPC for a nine-phase open-end winding (OW) permanent magnet synchronous machine, which is powered by nine H-bridge inverters with a common dc bus [28]. It should also be noted that the propagation of spiral circular waves in an excitable medium, in particular, in the heart muscle and nervous tissue, was considered in [29,30].

## 2. Evolution of Order Parameter and Steady State during HPT

As shown in [20,21], the Landau phenomenological theory satisfactorily describes the behavior of a material outside the elastic region of straining. This is due to the fact that in this theory a solid body is considered as a continuous medium. At the same time, it should be taken into account that under HPT, the dependence of the magnitude of the torsion torque on the number of revolutions can be approximated via a hyperbolic tangent. A more accurate dependence can be described within the framework of nonequilibrium evolutionary thermodynamics, where a connection is established between the processes of generation and annihilation of structural defects, as well as with materials hardening [16,31]. In this paper we will not investigate the dependence of the observed parameters on the torsion torque. Therefore, the corresponding quantity will not appear explicitly and will be included in the phenomenological constants. It should be noted that after a certain number of plunger revolutions, in a steady state, the dependence of properties on the torsion torque disappears, and therefore, the observed values will not change any more with increasing number of revolutions.

In a spatially homogeneous body, the value of the order parameter (OP) is determined by the minimum of the thermodynamic potential as a function of the OP, i.e., equality to zero of the first derivative of the potential with respect to OP. If this condition is not met, then a relaxation process occurs. As a result, the OP changes with time tending to an equilibrium value. In weakly nonequilibrium states (small values of the potential derivative with respect to the OP), the relaxation rate (derivative of the OP with respect to time) is also low. In Landau’s theory, which neglects OP fluctuations, it is assumed that the relationship between two derivatives is reduced to a simple proportionality. The corresponding equation was named Landau–Khalatnikov [32]. In a modified form, this equation can be written as
(1)$$\frac{\partial q_{i}}{\partial t}=-\gamma_{ij}\frac{\delta \Phi }{\delta q_{j}};\;\;\left(i=x,y\right)$$
where Φ is the free energy functional, $\frac{\delta \Phi }{\delta q}=\sum_{k}{\left(-1\right)}^{k}\frac{d^{k}}{dz^{k}}\frac{\partial \Phi }{\partial \left(\frac{\partial^{k}q}{\partial z^{k}}\right)}$ is the functional derivative, *t* is the time, and $\gamma_{ii}\left(i=x,y\right)$ is the matrix of kinetic coefficients characterizing the rate of relaxation of the system to the equilibrium state. *q_i_* here are the components of the vector order parameter of the system. It should be noted that in Equation (1), for simplicity, thermal noise is not taken into account. In what follows, we will assume that the kinetic coefficients $\gamma_{ii}\left(i=x,y\right)$ are constant, and for simplicity, we will neglect the cross effects between different components of the order parameter $\gamma_{xy}=\gamma_{yx}=0$. We will also introduce the order parameter being a combination of atomic displacements during a phase transition. In the experiment, this is reflected in a change in the interplanar distance in the crystal lattice and a change in its parameters [17,18,19].

The density of the nonequilibrium thermodynamic potential of a system in which an annular transient process occurs under the action of HPT with a moment directed along the OZ axis has the form
(2)$$\begin{array}{l} \Phi_{0}=\gamma_{1}\left({\left(\frac{\partial q\left(x,y\right)}{\partial x}\right)}^{2}+{\left(\frac{\partial q\left(x,y\right)}{\partial y}\right)}^{2}\right)+\frac{\alpha_{1}}{2}q^{2}\left(x,y\right)+\frac{\alpha_{2}}{4}q^{4}\left(x,y\right)+\frac{\alpha_{3}}{6}q^{6}\left(x,y\right)\\ \;\;\;\;\;+\overset{2}{\underset{i=1}{\sum }}\gamma_{2}(x,x+\left(-1{)}^{i}\Delta x,y\right)*q^{2}\left(x,y\right)*q^{2}(x+\left(-1{)}^{i}\Delta x,y\right)\\ \;\;\;\;\;+\overset{2}{\underset{i=1}{\sum }}\gamma_{2}(x,y,y+\left(-1{)}^{i}\Delta y\right)*q^{2}\left(x,y\right)*q^{2}(x,y+\left(-1{)}^{i}\Delta y\right) \end{array}$$
where $\alpha_{i}\;\left(i=1,2,3\right)$, $\gamma_{i}\;\left(i=1,2\right)$, are phenomenological parameters. The term $\gamma_{1}$ describes the inhomogeneities that arise during the passage of the wave in the XOY plane, the last two terms (the constant $\gamma_{2}$) determine the interactions with the environment in the nearest neighbor approximation (Δx≠0, Δy≠0). It should be noted that there are no higher invariants in potential (2). Their influence on the process will be discussed later. Substituting (2) into (1), after passing to polar coordinates (r, φ) we obtain the equation
(3)$$\begin{array}{l} \frac{\partial q}{\partial t}=-\gamma_{ii}(-\gamma_{1}(\frac{\partial^{2}q\left(r,\varphi \right)}{\partial r^{2}}+\frac{1}{r}\frac{\partial q\left(r,\varphi \right)}{\partial r}+\frac{1}{r^{2}}\frac{\partial^{2}q\left(r,\varphi \right)}{\partial \varphi^{2}}))+\alpha_{1}q\left(r,\varphi \right)+\alpha_{2}{\left(q\left(r,\varphi \right)\right)}^{3}+\alpha_{3}{\left(q\left(r,\varphi \right)\right)}^{5}\\ \;\;\;\;\;+2\overset{2}{\underset{i=1}{\sum }}\gamma_{2}(r,r+\left(-1{)}^{i}\Delta r,\varphi \right)*q\left(r,\varphi \right)*q^{2}(r+\left(-1{)}^{i}\Delta r,\varphi \right)\\ \;\;\;\;\;+2\overset{2}{\underset{i=1}{\sum }}\gamma_{2}(r,\varphi ,\varphi +\left(-1{)}^{i}\Delta \varphi \right)*q\left(r,\varphi \right)*q^{2}(r,\varphi ,\varphi +\left(-1{)}^{i}\Delta \varphi \right) \end{array}$$
where (Δr, Δφ) is the distance to the nearest neighbors in polar coordinates.

Equation (3) is a first order equation with respect to time t. It is known that if for equations of this type the condition of existence and uniqueness of the solution is met, then the phase trajectories do not intersect anywhere with a change in the initial conditions for each pair (r, φ), except for a limited number of isolated singular points. These points correspond to the stationary (steady) state of the system. In this state, all derivatives of the order parameter with respect to time are equal to zero. In our case, the singular point occurs at t→∞. Consequently, for a sufficiently long time interval after the application of HPT, a steady state should occur in the system.

## 3. Distribution of Order Parameter in the HPT Sample

Let at some point $\left(r_{0},\;\varphi_{0}\right)$ there be an anomalous value of the order parameter, which is different from the order parameter at other points. It may be due to the presence of a defect and is maintained constant under external influence. Let us study the distribution of the order parameter when applying HPT. It should be noted that, in a real situation, the order parameter of a defect also changes when applying HPT. However, in this problem, in the first approximation, this change will be neglected. The order parameter at points other than $\left(r_{0},\;\varphi_{0}\right)$ is determined from the algebraic equation obtained from (3) in the case of equality to zero of all derivatives both in time and in space. Since it is assumed that the outer boundary of the sample is not fixed, the unperturbed value of order parameter is one of the boundary conditions of Equation (3). This requirement is violated at the point $\left(r_{0},\;\varphi_{0}\right)$. The boundary condition at this point is the value of the corresponding order parameter at it. The second boundary condition is determined by the magnitude of the torsion torque for the HPT. Figure 1 shows the distribution of the order parameter when applying the HPT at the initial moment of time in the direction in which there is a defect.

Figure 2 shows the dynamics of changes in order parameter in the same direction at certain time intervals. It follows from Figure 2 that at the initial moments of time, the anomaly has practically no effect on the immediate environment. However, as we approach the steady state in the radial direction, a shelf appears on the curve due to the presence of an interaction between the nearest neighbors. A decrease in the absolute value of the constant $\gamma_{2}$ leads to an increase in the height and length of the shelf after the defect in the steady state, as well as the half-width of the hump in the region of the defect. Obviously, due to the small size of the defect, strong diffraction takes place and no shadow is formed.

Figure 3 shows the dynamics of the order parameter distribution for a small deviation in angle from the direction of the defect. Identical shading shows changes in the order parameter with a difference of half a degree (sampling length). The solid line shows the distribution of the order parameter along the direction of the defect, similar to the bottom graph in Figure 2. It can be seen that the distributions separated by half a degree differ in sign relative to the stationary line.

This is due to the presence of oscillations along the angle (Figure 4) in the vicinity of the point where the defect is located, which are due to the consideration of the tangential interaction between the nearest neighbors. In this case, the ratio of oscillation amplitudes increases relatively sharply with a change in the $\varphi_{0}$ angle. With a deviation of 3 degrees, the increase is 1.45. It should be noted that in Figure 3 and below the qualitative changes in OP are shown and all values are presented in arbitrary units. They depend on the value of the applied moment and the values of the interaction constants. 

Figure 4 shows the dynamics of changes in the oscillatory process depending on the value of the polar radius. It can be seen that the maximum oscillations are observed in the area where the defect is located. For larger or smaller values of the modulus of the polar vector $\gamma_{3}\left(r,\;\varphi ,\;\varphi \pm \Delta \varphi \right)$, a strong attenuation of this process is observed. There are no fluctuations at the outer boundary. It should be noted that a shift in the position of the defect closer to the boundary and an increase in the magnitude of th3e radial interaction at the boundary can also lead to slight fluctuations. A decrease in the modulus leads to a decrease in the width of the oscillatory process both in angle and in radius.

Figure 5 shows the dynamics of changes in the oscillatory process at a point with an angle coinciding with the defect angle and a radius vector smaller than the corresponding defect vector. The upper graph corresponds to the initial value of time, and the lower graph corresponds to the final one (steady state).

## 4. Influence of the Higher-Order Invariants

Potential (2) takes into account the change in the order parameter with temperature and pressure, as well as the appearance of inhomogeneity when applying HPT. However, various kinds of interactions take place in matter, which are described by higher-order invariants. One of the types of interaction can be represented as the product of the order parameter and its derivative. Thus, the new nonequilibrium thermodynamic potential has the form
(4)$$\Phi =\Phi_{0}+\Phi_{1}$$
where $\Phi_{0}$ is given by expression (2) and
(5)$$\Phi_{1}=\gamma_{3}q^{2}\left(r,\;\varphi \right)\;{\left(\frac{\partial q\left(r,\;\varphi \right)}{\partial r}\right)}^{2}+\gamma_{4}q^{2}\left(r,\;\varphi \right){\left(\frac{\partial q\left(r,\;\varphi \right)}{\partial \varphi }\right)}^{2}$$

At $\gamma_{4}=0,\gamma_{3}<0$, the distribution of the order parameter along the direction of the defect is slightly deformed with respect to the case $\gamma_{4}=0,\gamma_{3}=0$ (see Figure 3). The half-width of the peak in the region of the defect decreases, and it becomes sharper. All curves on the graph take less of a “pear shape”. In this case, the range of changes along the radius in the region of the defect also decreases. When the peak begins to blur at $\gamma_{4}=0,\gamma_{3}>0$, the curves in the graph in Figure 3 acquire a more “pear-shaped” shape. The range of order parameter variation along the radius in the vicinity of the defect point also increases.

If $\gamma_{4}>0,\gamma_{3}=0$, then the corresponding rings in Figure 3 are compressed, and consequently, the range and the region of oscillations along the radius decrease in comparison with Figure 4. At $\gamma_{4}<0,\gamma_{3}=0$, the rings expand and the region of oscillations along the radius increases. Accordingly, the range of oscillations increases. Hence, we can conclude that taking into account higher invariants leads only to quantitative changes in the distribution of the order parameter and does not give qualitatively new results. In conclusion, it should be noted that there are currently no corresponding experimental works, but the results obtained can be used in their planning.

## 5. Conclusions

The model based on the Landau phenomenological theory and Landau–Khalatnikov equation was developed to describe behavior of a material outside the elastic region of straining. The order parameter is introduced in the model, being a combination of atomic displacements during a phase transition. In the experiment, this is reflected in a change in the interplanar distance in the crystal lattice and a change in its parameters. The distribution of the order parameter in the HPT sample is analyzed with increasing duration of the HPT treatment. The transition to the steady state after a certain time is predicted. The influence of higher-order invariants on the possible oscillations of the order parameter is also analyzed.

## Figures and Tables

**Figure 1 materials-15-06970-f001:**
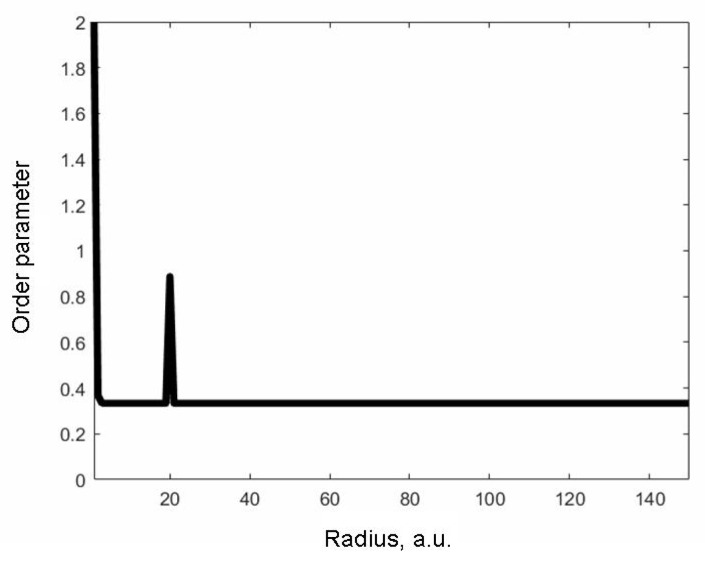
Distribution of order parameter at the initial moment of time in the direction of the defect. The values along the axes are given in relative units.

**Figure 2 materials-15-06970-f002:**
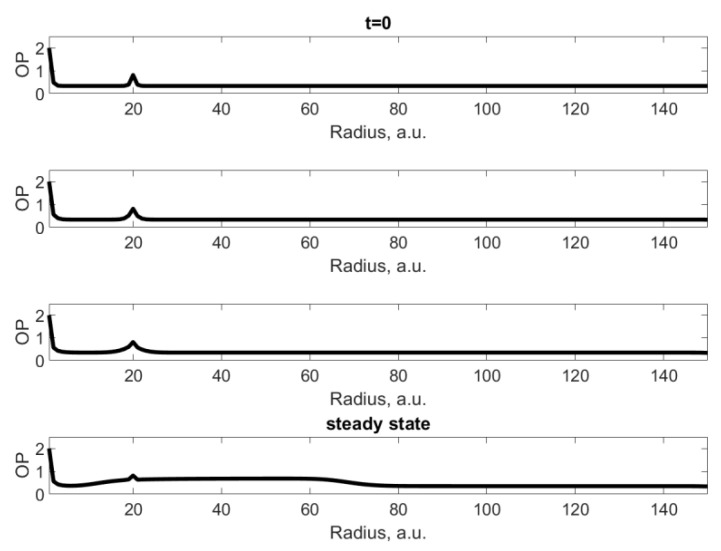
Dynamics of order parameter distribution depending on time (increasing from top to the bottom) in the direction of the defect. The bottom plot corresponds to the steady state.

**Figure 3 materials-15-06970-f003:**
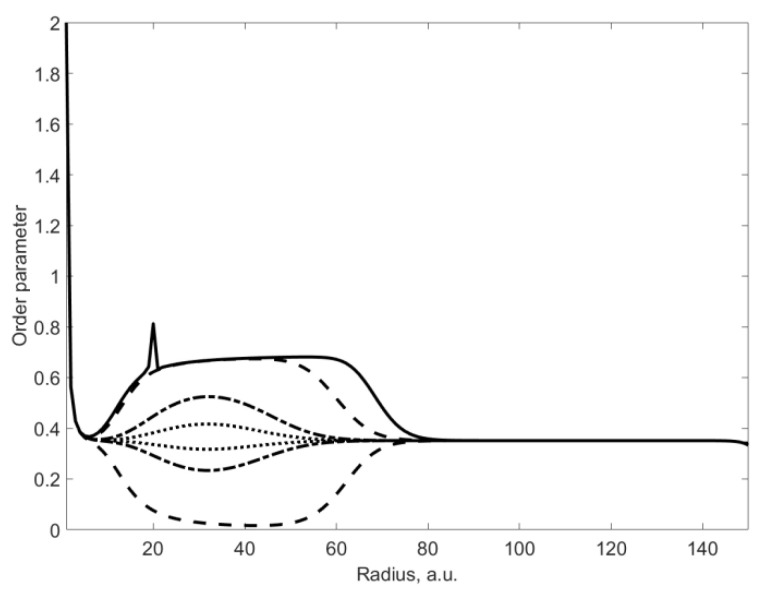
Dynamics of the order parameter distribution depending on the angle in the vicinity of the defect.

**Figure 4 materials-15-06970-f004:**
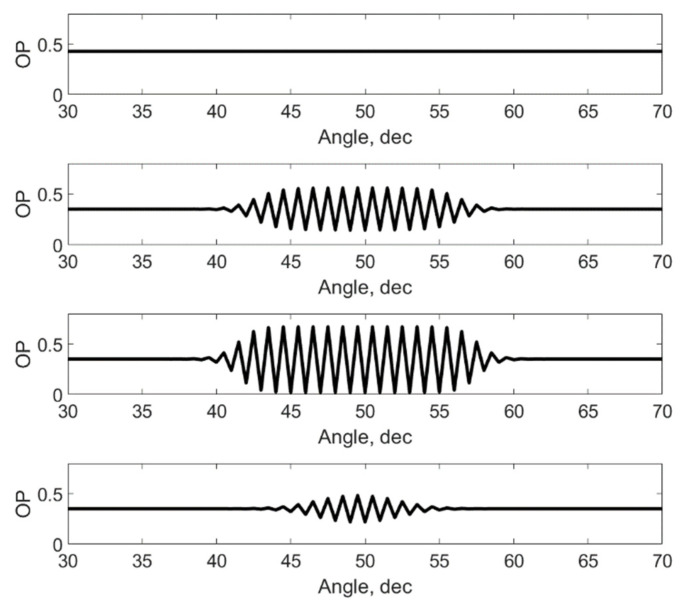
Dynamics of the order parameter distribution depending on the radius in the vicinity of the defect.

**Figure 5 materials-15-06970-f005:**
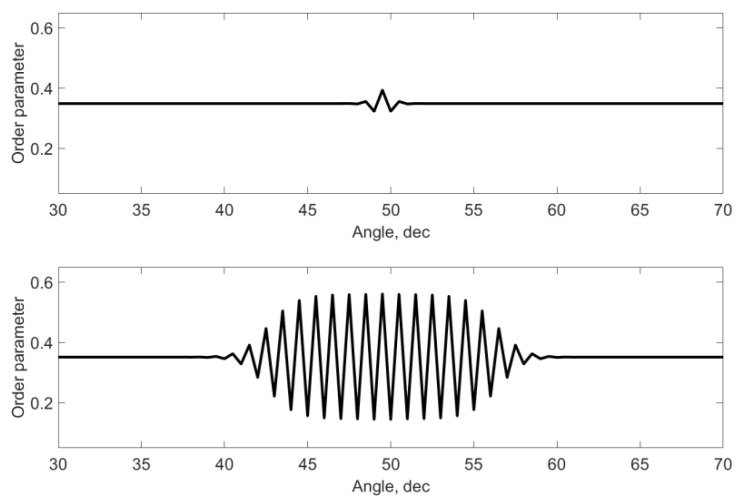
Dynamics of changes in the oscillatory process in time.

## Data Availability

Not applicable.

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
