# Peer review of "Distribution of Order Parameter in Solids under High Pressure Torsion"

_materials, 2022, doi:10.3390/ma15196970_

Round 1

Reviewer 1 Report

The problem of the radial distribution of a linear combination of the displacement of the unit cell atoms without fixing the outer boundary of the sample has not yet been explored. This paper investigates this problem in the flawed case. In particular, the propagation of spiral circular waves in the excitable media was considered. This paper considers the actual situation, which has an important reference significance for practical applications. This study is very interesting and meaningful.

The main questions are as follows:

1. This paper uses the Landau-Khalatnikov equation to solve the deviation problem, but can the formula solve it completely? What is the principle of exporting the formula?

2. Please explain the basis of the fluctuation range in Figure 3.

3. Are the simulation results compared with the test results? Test results shall be added.

Author Response

Reviewer 1

The problem of the radial distribution of a linear combination of the displacement of the unit cell atoms without fixing the outer boundary of the sample has not yet been explored. This paper investigates this problem in the flawed case. In particular, the propagation of spiral circular waves in the excitable media was considered. This paper considers the actual situation, which has an important reference significance for practical applications. This study is very interesting and meaningful.

The main questions are as follows:

  1. This paper uses the Landau-Khalatnikov equation to solve the deviation problem, but can the formula solve it completely? What is the principle of exporting the formula?

In a spatially homogeneous body, the value of the order parameter (OP) is determined by the minimum of the thermodynamic potential as a function of the OP, i.e. equality to zero of the first derivative of the potential with respect to OP. If this condition is not met, then a relaxation process occurs. As a result the OP changes with time tending to an equilibrium value. In weakly nonequilibrium states (small values of the potential derivative with respect to the OP), the relaxation rate (derivative of the OP with respect to time) is also low. In Landau's theory, which neglects OP fluctuations, it is assumed that the relationship between two derivatives is reduced to a simple proportionality. The corresponding equation was named Landau-Khalatnikov [32].

  1. Please explain the basis of the fluctuation range in Figure 3.

The ratio of the oscillation amplitudes increases relatively sharply when the  angle changes. With a deviation of 3 degrees, the increase is 1.45. It should be noted that in Fig. 3 and below, qualitative changes in order parameter are shown and all values are presented in relative units. They depend on the value of the applied moment and the values of the interaction constants.

  1. Are the simulation results compared with the test results? Test results shall be added.

Unfortunately, there are currently no corresponding experimental works. Therefore, there is no comparison. However, the results obtained can be used in the planning of experimental works.

The explanations for formulae were corrected

Author Response

Reviewer 2

Reviewer comments on the paper “Distribution of Order Parameter in Solids under

High Pressure Torsion”

In this paper, high pressure torsion is described to show characteristics of phase transformation and severe plastic deformation. The paper in this stage is incomplete and cannot be accepted. The results are very briefly organized. A revised paper may change my decision. The reviewer comments are suggested as:

Although the title, materials and methods of the present paper are very interesting, however the main novelties of it are not clear for the reviewer. Authors are encouraged to add more comments on the novelties and main contributions of the present paper in Abstract and last paragraph of Introduction section.

We added more comments on the novelties and main contributions of the present paper in Abstract and last paragraph of Introduction section.

What is application of the proposed method? Authors are suggested to provide some technical expressions on the application of the proposed method and needing to this new finding.

The Introduction section is very briefly organized. The Introduction section should be improved using the papers on the subjected of the paper such as:

Front. Energy Res. 2022, 10:937299;

J Math Chemistry, 2022, 60, 475–501;

Sustainable Energy Techn Assess, 2022, 53, 102438;

Int J Elect Power & Energy Syst, 2022, 141, 108114;

IEEE trans indust electr (1982), 2022, 69(6), 5386-5397.

The Introduction section was improved using the suggested papers on the subject:

This problem arises when studying the distribution of stresses during the acceleration of the fan blades, centrifugal pump, etc. In particular, it was shown in [23] that when the blade of a centrifugal pump rotates, the pressure on the surface of the magnetic fluid sealing film has a periodic pulsation. This pressure is transferred to the blades, which leads to a change in the distribution of stress and order parameter (OP) on them. Also, this issue is relevant in the study of voltage distribution in power networks in the event of peak loads [24,25]. In order to ensure the emission reduction in the process of extracting energy from fuel one carryies out the reactions far from stoichiometric conditions. The practical implementation of this requires a search for a compromise between emission, combustion stability and efficiency and is associated with the swirling of the flame and the use of vortex technologies based on the novel algorythms [26]. For example, the finite control set model predictive control strategy (FCS-MPC) with fast response and a simple and flexible structure was used for the first time investigates the FCS-MPC for a nine-phase open-end winding (OW) permanent magnet synchronous machine, which is powered by nine H-bridge inverters with a common dc bus [27]. It should also be noted that the propagation of spiral circular waves in an excitable medium, in particular, in the heart muscle and nervous tissue, was considered in [28, 29].

All variables and component should be defined after first appearance in the paper.

We defined all variables and components after first appearance in the paper

All equations and relations are presented and used without any description and explanation.

We added some explanations in particular:

“In a spatially homogeneous body, the value of the order parameter (OP) is determined by the minimum of the thermodynamic potential as a function of the OP, i.e. equality to zero of the first derivative of the potential with respect to OP. If this condition is not met, then a relaxation process occurs. As a result the OP changes with time tending to an equilibrium value. In weakly nonequilibrium states (small values of the potential derivative with respect to the OP), the relaxation rate (derivative of the OP with respect to time) is also low. In Landau's theory, which neglects OP fluctuations, it is assumed that the relationship between two derivatives is reduced to a simple proportionality. The corresponding equation was named Landau-Khalatnikov [32]. In a modified form, this equation can be written as (1)”

Authors are encouraged to add the relevant references for citation of the basic equations and relations.

We added the relevant references for citation of the basic equations and relations.

The discussion is not informative. It should be enriched with addition of some more important conclusions. The results and discussion are very briefly organized. An extended result and discussion is required.

We extended the results and discussion

Round 2

Reviewer 2 Report

The authors provided a detailed revised paper based on reviewer comments.

The paper is suggested for publication.